# Dysregulated Epicardial Adipose Tissue as a Risk Factor and Potential Therapeutic Target of Heart Failure with Preserved Ejection Fraction in Diabetes

**DOI:** 10.3390/biom12020176

**Published:** 2022-01-21

**Authors:** Teresa Salvatore, Raffaele Galiero, Alfredo Caturano, Erica Vetrano, Luca Rinaldi, Francesca Coviello, Anna Di Martino, Gaetana Albanese, Sara Colantuoni, Giulia Medicamento, Raffaele Marfella, Celestino Sardu, Ferdinando Carlo Sasso

**Affiliations:** 1Department of Precision Medicine, University of Campania Luigi Vanvitelli, Via De Crecchio 7, I-80138 Naples, Italy; teresa.salvatore@unicampania.it; 2Department of Advanced Medical and Surgical Sciences, University of Campania Luigi Vanvitelli, Piazza Luigi Miraglia 2, I-80138 Naples, Italy; raffaele.galiero@unicampania.it (R.G.); alfredo.caturano@unicampania.it (A.C.); erica.vetrano@unicampania.it (E.V.); luca.rinaldi@unicampania.it (L.R.); francesca.coviello@studenti.unicampania.it (F.C.); annadimarti@alice.it (A.D.M.); gaetanaalbanese@hotmail.it (G.A.); sara.colantuoni@yahoo.it (S.C.); med.giulia@gmail.com (G.M.); raffaele.marfella@unicampania.it (R.M.); celestino.sardu@unicampania.it (C.S.); 3Mediterrannea Cardiocentro, I-80138 Naples, Italy

**Keywords:** type 2 diabetes, epicardial fat, heart failure

## Abstract

Cardiovascular (CV) disease and heart failure (HF) are the leading cause of mortality in type 2 diabetes (T2DM), a metabolic disease which represents a fast-growing health challenge worldwide. Specifically, T2DM induces a cluster of systemic metabolic and non-metabolic signaling which may promote myocardium derangements such as inflammation, fibrosis, and myocyte stiffness, which represent the hallmarks of heart failure with preserved ejection fraction (HFpEF). On the other hand, several observational studies have reported that patients with T2DM have an abnormally enlarged and biologically transformed epicardial adipose tissue (EAT) compared with non-diabetic controls. This expanded EAT not only causes a mechanical constriction of the diastolic filling but is also a source of pro-inflammatory mediators capable of causing inflammation, microcirculatory dysfunction and fibrosis of the underlying myocardium, thus impairing the relaxability of the left ventricle and increasing its filling pressure. In addition to representing a potential CV risk factor, emerging evidence shows that EAT may guide the therapeutic decision in diabetic patients as drugs such as metformin, glucagon-like peptide-1 (GLP-1) receptor agonists and sodium-glucose cotransporter 2 inhibitors (SGLT2-Is), have been associated with attenuation of EAT enlargement.

## 1. Introduction

Diabetes mellitus represents a critical health care problem with a worldwide escalation mostly related to ageing population, socio-economic development, unhealthy diet regimes, and sedentary lifestyle. Its global estimated prevalence was 9.3% in 2019 and is projected to cover a quarter of the global population in 2030, and more than half in 2045 [1]. As a result, a substantial disease burden is expected, mainly due to CV complications that represent the leading cause of death among diabetic people, although the morbidity of the other chronic complications of diabetes overall has a high prevalence and a huge impact on National Health Systems [2,3,4,5,6,7].

Since the report of the Framingham Heart study, type 2 diabetes (T2DM) has been known to increase the incidence of HF up to more than double with respect to non-affected individuals, with the growing awareness that this complication may develop independently of coronary artery disease (CAD) and associated risk factors [8,9,10,11]. A higher risk of hospitalization for HF among T2DM patients was reported in a large cohort study, even with CV risk factors within target ranges [12]. Otherwise, in a prospective study in patients with type 1 diabetes (T1DM) followed over 7 years, HF occurred only when diabetes coexisted with hypertension and CAD, the two main CV disorders affecting myocardial function [13]. Diabetes markedly increases the development, morbidity and mortality of HF with preserved ejection fraction (HFpEF), showing a particularly close link with this clinical entity [14,15]. HFpEF is a complex and heterogeneous syndrome that, despite its growing prevalence and incidence relative to HF with reduced ejection fraction (HFrEF), even retains important knowledge gaps and therapeutic challenges [16,17]. It is likely that a better understanding of its pathobiology may make it possible to identify effective treatment modalities and thereby ameliorate its prognosis. In recent years, the excess of epicardial adipose tissue (EAT) ended up in the spotlight as a main actor in CV disease and as a detrimental factor promoting inflammation, microvasculopathy, fibrosis, and stiffness of myocardium, all findings corresponding to the hallmarks of HFpEF [18]. On the other hand, diabetic patients often exhibit an abnormally expanded EAT, usually in association with obesity and metabolic syndrome in both T2DM and T1DM [19,20,21,22].

In this review, we want to highlight the available evidence indicating that EAT may act as a possible intermediary coupling diabetes and HFpEF, and that it may positively respond to fat targeted therapies. The aim would be to emphasize the utility of EAT, considered as an additional cardiometabolic diagnostic marker in the field of assessment of diabetes complications, and as an adjunctive therapeutic target given its fast responsiveness to glucose-lowering drugs.

## 2. Anatomical Features and Functional Properties of EAT

### 2.1. Embryonic Origin and Anatomy

The adipose tissue surrounding the heart involves two types of depots, the pericardial adipose tissue (PAT) and the EAT, even if the nomenclature is confusing, as some researchers use the terms paracardial or intrathoracic or mediastinal for pericardial fat, and the term pericardial fat to indicate the combination of paracardial fat with EAT [23]. PAT rests on the external surface of the pericardial fibrous layer, consists of adipocytes originating from the primitive thoracic mesenchyme, and is vascularized by non-coronary arteries. Otherwise, EAT is located under the visceral pericardium in anatomical continuity with the myocardium and is embriologically derived, as mesenteric and omental fat, from the brown adipose tissue (BAT) of splanchnic mesoderm. Importantly, EAT is supplied by branches of coronary arteries with a shared microcirculation that facilitate a direct crosstalk with the myocardium [24]. Based on these features, only EAT may be considered to be a true heart-specific visceral fat [25,26]. In healthy individuals, EAT accounts for ~20% of total heart weight and is heterogeneously distributed on an area exceeding 80% of the cardiac total surface. It mainly occupies the atrioventricular and interventricular grooves and surrounds the right coronary artery and the left anterior descending coronary artery, and to a lesser extent covers atria, free wall of right ventricle, and apex [25,26]. Microscopically, EAT is mainly composed of white adipocytes specialized in energy storage, with a higher cellularity than subcutaneous and other visceral fat depots, with a prevalence of preadipocytes over mature adipocytes [27,28].

### 2.2. Functional Properties

EAT may contribute to heart physiology by mechanical, metabolic, thermogenic, and paracrine/vasocrine mechanisms. Due to its spatial distribution, EAT exerts mechanical protection of coronary arteries from excessive distortion and compression during the contraction of neighbouring myocardium [29,30]. Compared with subcutaneous adipose tissue (SAT), human EAT is richer in saturated fatty acids (FAs) and poorer in unsaturated ones [31]. This different composition may account for its greater capacity of mobilization, deposition, and synthesis of FAs than all body fat and other visceral adipose depots, as observed in adult guinea pigs [32]. Thanks to this remarkable flexibility of FA turnover, EAT can play the important physiological role of metabolic sensor for heart, which greatly depends on FA oxidation as primary fuel [33]. To that end, EAT provides FAs directly to myocardium at times of high energy demand and implements the local triacylglycerol storage as request decreases [34,35]. In addition, the high rates of lipogenesis displayed by EAT act as scavenger for excess FAs in systemic circulation to protect against their myocardial lipotoxicity responsible of functional deterioration and possible lethal arrhythmias [36]. Animal and human data indicate that EAT is phenotypically brown during the early stages of life. Despite whitening with age, it retains in adulthood the biological property to combust proinflammatory FAs, showing molecular features and a gene profile of beige adipocytes, a new class of adipocytes that display the properties of brown adipocytes, but are located within WAT depots [24,37]. In a study on fat samples taken at open heart surgery, the expression of uncoupling protein-1 (UCP-1), the inner mitochondrial membrane protein that is a specific marker for and a mediator of nonshivering thermogenesis in brown adipocytes, was 5-fold higher in epicardial than substernal fat and basically undetectable in SAT [38]. The chronic exposure to cold promotes the activation of epicardial fat peroxisome proliferator-activated receptor γ (PPAR-γ) coactivator 1-α (PGC 1-α), a key mediator of the white-to-beige adipocyte transformation [39]. These BAT-like features of EAT could defend myocardium against hypothermia or unfavorable hemodynamic conditions, such ischemia or hypoxia. However, there is no direct evidence of heat production, and a role in regulation of myocardial and/or vascular redox state has been suggested [40]. EAT is an extremely active endocrine organ producing adipocytokines, not only ones involved in thermogenesis and regulation of lipid and glucose metabolism, but even those capable of both pro-inflammatory and anti-inflammatory response (interleukin (IL)-1b, interleukin (IL) 6 and interleukin (IL) 6 soluble receptor, tumor necrosis factor-α (TNF), adiponectin). Through the shared microcirculation, these bioactive molecules may locally modulate the structure and function of adjacent myocardium in a paracrine fashion [25]. There is even evidence for a microcirculatory connection between EAT and the coronary wall, by which cytokines may be released from epicardial tissue directly into vasa vasorum and transported downstream into the arterial wall by mean of a vasocrine-signaling mechanism [30]. Thus, an interrelation of pericoronary fat with vasomotor function is mediated by the release of specific vasoactive factors such as leptin and adiponectin among others [41,42]. EAT has a unique transcriptome enriched in genes involved in coagulation, endothelial function, phospholipase activity, apoptosis, and immune signaling, and a secretome that may be influenced by environmental, epigenetic, and genetic factors [43,44,45]. Under physiological conditions, EAT secretes the adiponectin, an adipocyte-derived adaptive adipokine that nourishes the myocardium contributing to FAs combustion, and exerts a series of protective actions against hypertrophy stimuli for cardiomyocytes and against inflammation and fibrosis of myocardium and coronary arteries [18]. Other anti-inflammation adipokines may be secreted by a normal EAT such as adrenomedullin, C1q/TNF-related proteins, omentin, and secreted frizzled-related protein 5 [46,47]. Additionally, EAT is a major source of mesenchymal stem cells that, transiting through the shared microcirculation, can provide the regeneration and repair of injured myocardium [48,49]. As opposed to these protective actions, detrimental roles for heart are attributed to epicardial fat [50]. An imbalanced proinflammatory secretome promoting the EAT synthesis of leptin, TNF-α, IL-1b and IL-6, may disrupt myocardial function. These results have been found in a small population of patients with CAD and the local inflammatory burden was independent of other clinical variables and of the plasma concentrations of circulating cytokines. Moreover, in the epicardial fat of a sub-group of patients, the authors observed an up-regulation of almost 800 inflammatory and immune response genes [51]. Similarly, a high-FA feeding of the heart by EAT may lead to intramyocardial fat accumulation causing functional derangements [52]. This is what can happen when an enlarged and biologically transformed EAT coats the heart, thus losing its physiological protective role, and acquiring lipotoxic effects throughout an excess FFa release, which leads to abnormal lipid deposition and fatty infiltration in the myocardium [52].

## 3. Epicardial Adipose Tissue in Diabetes

An expansion of EAT is associated with insulin resistance and visceral fat accumulation, the core defects of metabolic syndrome and two fundamental elements of T2DM that frequently precede its diagnosis by a long-lasting period [53]. It has been calculated that EAT thickness is almost double in individuals with metabolic syndrome compared to controls, with a significant correlation with each component of syndrome [54]. In non-diabetic people, impaired fasting glucose, or obesity-related reduction of insulin sensitivity, as assessed by euglycemic hyperinsulinemic clamp and other markers of insulin resistance, are associated with increased epicardial fat [19,55]. Various studies indicate that an enlarged EAT mass in T2DM is associated with obesity and metabolic syndrome and its components [21,56]. In T1DM, a study describes a higher epicardial fat independently of obesity [57]. Instead, in a pilot study in T1DM patients from the DCCT/EDIC (Diabetes Control and Complications Trial/Epidemiology of Diabetes Interventions and Complications), the epicardial accumulation of adipose tissue was strongly associated with elements of metabolic syndrome such as higher body mass index (BMI) and waist-to-hip ratio, and elevated triglycerides [58]. A recent meta-analysis including 13 relevant studies for a total of 1102 diabetic patients and 813 healthy subjects, suggests that the amount of EAT is significantly higher in individuals with diabetes than healthy controls, irrespective of T1DM or T2DM, BMI and EAT measurement techniques [59]. Importantly, EAT adiposity is associated with alterations of myocardial function in T2DM people and may be a predictor of diastolic dysfunction in new diagnosed patients [60,61]. Already two decades ago, EAT was identified as a source of inflammatory mediators in high-risk cardiac patients [51]. In a bioptical study comparing SAT and EAT among subjects with or without diabetes, adipocyte size in EAT was significantly larger in diabetic than non-diabetic subjects, and both SAT and EAT were featured in diabetic group by a predominantly inflammatory profile [62]. Recent genetic investigations evidence that the EAT transcriptome of T2DM subjects is highly enriched in immune genes like Pentraxin 3 and endothelial lipase G compared to SAT from the same individuals [44]. An altered secretory profile associated with increased volume of EAT is confirmed in a population of T2DM patients with CAD [63].

## 4. Diabetes and HFpEF

### 4.1. Pathophysiological Mechanisms

The major causes of HF in diabetes include CAD and hypertension, but a direct detrimental effect of diabetes itself on myocardium may greatly contribute. Both experimental and clinical evidence state that diabetic individuals are predisposed to a distinct cardiomyopathy, named diabetic cardiomyopathy, a heart muscle-specific disease characterized by abnormalities of myocardial structure and performance in the absence of other cardiac risk factors, such as CAD, hypertension, and significant valvular disease. Several pathophysiological mechanisms associated with diabetes may contribute to this entity, namely oxidative stress, inflammation, impaired mitochondrial energetics, altered intracellular calcium handling, and increased neuro-humoral activation, that all together determine myocardial inflammation, hypertrophy, fibrosis, and coronary microvascular disease [64]. Diabetic cardiomyopathy may clinically manifest as a condition of eccentric left ventricle (LV) dilatation impairing systolic function, as classically described, or as a restrictive concentric LV remodeling determining diastolic dysfunction. These two phenotypes are not successive stages of diabetic cardiomyopathy, but two pictures that independently may evolve to HFrEF or HFpEF, respectively [65]. The pathophysiological mechanisms underlying myocardial remodeling seem to be differential and to mainly consist in the coronary microvascular endothelial dysfunction for HFpEF and cardiomyocyte cell death for HFrEF [66].

According to a paradigm suggested by Paulus and Tschöpe almost a decade ago, HFpEF may be considered an inflammatory cardiometabolic disease in which comorbidities involving a systemic inflammatory condition induce oxidative stress and endothelial dysfunction in the coronary microvascular endothelium. It seems that hyperglycemia per se can directly or indirectly affect the cardiomyocytes activity. Directly, hyperglycemia can determine mitochondrial network fragmentation and oxidative stress, with a consequent contractile dysfunction [65]. Indirectly, through a paracrine endothelial signaling, hyperglycemia causes a mitochondrial hyper-production of superoxide, with impaired activation of endothelial nitric oxide synthase. As a result, the impaired bioavailability in nitric oxide impairs its downstream cGMP protein kinase G (PKG) pathway in cardiomyocytes which become hypertrophied and stiffened [65,67]. In line with this pathophysiological mechanism, a past bioptic study in diabetic patients demonstrated that a raised LV diastolic stiffness was peculiar of HFpEF, predominantly due to the increased resting tension of hypertrophied cardiomyocytes since a deficit in phosphorylation of the N2B region of titin by protein kinase A. Otherwise, fibrosis and advanced glycation end products (AGEs) were more importantly represented when left ventricular ejection fraction (LVEF) was reduced [68]. In an autopsy study in patients with ante mortem diagnosis of HFpEF, the myocardium morphology was featured by severe hypertrophy, microvascular rarefaction, and fibrosis [69]. The last two alterations are likely triggered by microvascular endothelial inflammation [70].

### 4.2. Phenotype and Main Heart Dysfunctions

HFpEF is a heterogeneous syndrome characterized by a frequent association with multiple comorbidities (mainly diabetes and obesity), a complex pathophysiology originating from the interplay of cardiac and extracardiac abnormalities, and a consequent variegate clinical expression with different phenotypes [71]. In a recent study in participants of TOPCAT (Treatment of Preserved Cardiac Function Heart Failure with an Aldosterone Antagonist Trial), three phenogroups were identified, one of which represented by “Obese, diabetic with advanced symptoms”. These subjects exhibited the worst overall prognosis and the highest levels of biomarkers of TNF-α mediated inflammation [72]. In the past decade, HFpEF has taken on great importance in individuals with diabetes who represent about 45% of people suffering from this type of HF, with a significantly increasing proportion in those with new-onset HFpEF [73]. Thirty percent of HFpEF patients have both obesity and T2DM, and this prevalence is expected to grow worldwide due to the pandemic of these diseases, as well as the longer life expectancy [74,75]. Both morbidity and long-term mortality of HFpEF are greatly worsened by a coexisting diabetes and this adverse outcome cannot be offset by effective treatments [71,76]. The poor prognosis of HFpEF in diabetic patients may depend on a dangerous CV phenotype characterized by impaired chronotropic reserve and marked LV hypertrophy, associated with increased biomarkers indicating activation of multiple deleterious pathways of fibrosis, oxidative stress, and inflammation [77]. Lam outlined the importance of considering diabetes as a condition of risk for HFpEF that may evolve over time in asymptomatic LV diastolic dysfunction and finally in the symptomatic stage of HFpEF [78]. A diastolic dysfunction can develop early in T2DM course and be detected in 75% of asymptomatic patients, with a prevalence and a severity that increase across the whole spectrum of glucose dysregulation from prediabetes to diabetes [79,80]. Furthermore, the diastolic dysfunction is a strong predictor of incident HF and CV mortality in T2DM [81,82]. These findings outline the importance of implementing preventive measures in HFpEF diabetic patients.

## 5. Role of EAT Expansion in the Pathophysiology of HFpEF in Diabetes

An enlarged EAT may impair cardiac function through various mechanisms. Basically, it may concur to the development of a CAD-related HF, because the excess of fat surrounding the coronary arteries causes inflammation and accelerated atherosclerosis at this level, acting in a paracrine manner. This hypothesis is supported by the observation that EAT thickness was significantly correlated with the presence and severity of CAD and corroborated by the arrest of CAD progression after experimental resection of the epicardium [83,84]. The role of EAT expansion in the pathophysiology of HFpEF in diabetic patients is synthesized in Figure 1.

### Mechanical and Metabolic Mechanisms

Epicardial fat interacts directly with the heart both mechanically and metabolically. Occupying a fixed pericardial space, the EAT mass may drive an abnormal ventricular distensibility determining a constrictive pattern responsible of pressure equalization across ventricles in different physiologic states, particularly during exercise and in patients with obesity and with or without diabetes [85,86].

In systemic disorders such as T2DM and obesity, the adipocytes within the epicardium take on the features of white adipose tissue (WAT), namely they are overfilled with lipids and prone to lipolysis, releasing an excess of FAs [52]. The consequent heart overfeeding may promote a myocardial steatosis that has been associated with the development of ventricular dysfunction [87]. In studies using proton magnetic resonance spectroscopy, EAT expansion was associated with intramyocardial accumulation of triglycerides in severely obese subjects and intramyocardial fat was correlated with LV diastolic dysfunction [88]. Moreover, in a population of non-obese patients, with or without diabetes, fat deposition in myocardium was significantly more represented in subjects with HFpEF than HFrEF or non-HF controls [89]. More interestingly, according to a suggestive and convincing pathophysiological reconstruction formulated by some experts, an EAT enlarged and biologically transformed by diabetes and/or obesity, may transfer intrinsic injury towards the heart already damaged by the same systemic metabolic disorders. In other words, the anatomic proximity of EAT to the myocardium may characterize this structure as a transducer that address and amplify systemic derangements onto the heart through paracrine and vasocrine mechanisms [18,90]. This effect can be realized because there is no muscle fascia between EAT and myocardium, and the two tissues are intimately connected through an unobstructed shared microcirculation.

In T2DM and obesity, the adipocytes of expanded EAT release large amounts of FAs that trigger macrophage infiltration and secretion of pro-inflammatory adipocytokines including leptin, TNF-α, IL-6, IL-1b and resistin. Simultaneously, the production of adiponectin diminishes [18,91]. These biological derangements may be transmitted locally to the neighboring cardiac muscle. Such a scenario is strongly suggested by the study of Greulich et al. indicating that cardiomyocytes of rat incubated with conditioned media generated from explants of EAT biopsies performed in T2DM patients became dysfunctional and insulin resistant [92]. The study also demonstrated that the secretion pattern of EAT was considerably different from SAT and PAT, and in diabetic compared to non-diabetic individuals. As a consequence, the myocardial damage induced by systemic inflammation is perpetrated by a pathologic EAT that adversely affects structure and function intensifying cardiac inflammation, coronary microvascular dysfunction, and pro-fibrotic mechanisms, the pathologic hallmarks of HFpEF [93]. Among inflammatory cytokines, leptin can play a relevant role as its circulating levels were correlated with EAT adiposity in T1DM patients independently of obesity, and in a study in CAD patients both leptin gene (AA genotype) and leptin receptor gene (RR genotype) carried a risk that was at least tripled of developing HFpEF [57,94]. Moreover, leptin adversely affects calcium handling in cardiomyocytes so as to impair myocardial relaxation, stimulates the synthesis of collagen inducing cardiac fibrosis by increasing aldosterone secretion, and enhances the sodium tubular reabsorption [95,96,97]. As evidence, in patients with hypertension and/or T1DM, an impaired coronary flow reserve has been associated with diastolic dysfunction, and, more recently, with amount of ventricular EAT [98,99]. A study in non-obese, with or without T1DM/T2DM patients without obstructive narrowing of the left anterior descending artery, investigated the coronary microcirculation through multi-detector computed tomography and demonstrated that periventricular EAT accumulation was significantly associated with impaired coronary flow reserve (CFR) and deterioration of LV diastolic function. Since CFR depends on the combined effects of microvascular dysfunction and epicardial coronary stenosis that was absent in the studied patients, the impaired CFR likely reflected the presence of a microvasculopathy [100]. Both inflammation and microvasculopathy promote cardiac fibrosis with the contribution of mesenchymal stem cells of the epicardium that can migrate into myocardium, where they can mature into fibroblasts [101]. The final result is a decreased ventricular compliance which presents with impairment in relaxation or increase in stiffness of LV. Even if the diastolic dysfunction is the hemodynamic feature of HFpEF, the essence of its pathophysiology is the increase of LV filling pressure, meaning that the heart can adequately pump the blood at expense of an elevation of cardiac filling pressures that, if disproportionate during exercise or at rest, are responsible of dyspnea [102,103]. Clinical observations strongly support the pathophysiological process which starts from EAT expansion and inflammation and results in HFpEF. A total EAT volume significantly higher, particularly in T2DM patients, was found in HF patients with mid-range and preserved EF when compared to well-matched controls with similar BMI. In addition, epicardial fat was positively associated with biomarkers of myocardial damage such as creatine kinase-MB, troponin T [104]. Cardiac magnetic resonance (CMR) imaging revealed more prevalent fibrosis in HFpEF subjects than in control subjects, and the first group was associated with adverse outcomes [105,106].

## 6. Diagnosis of EAT Enlargement

The great clinical implication associated with epicardial fat resides in its measurability by non-invasive imaging techniques which may allow its use as a tool for prediction of cardiometabolic risk and a modifiable therapeutic target [107]. The simplest method to evaluate EAT is the measurement of the hyperechoic area between epicardial surface and parietal pericardium on the free wall of the right ventricle by transthoracic two-dimensional echocardiography [108]. While rapid, relatively inexpensive, and widely available, this technique is limited by a high dependency on operator expertise and a linear measurement at a single location, which fails to take account for the site variability of epicardial fat thickness [109]. Moreover, echocardiography detects EAT thickness rather than volume, a parameter that can be provided by the three-dimensional imagines of cardiac computed tomography (CCT) and CMR. The CCT technique enables an accurate and reproducible quantification of EAT, even in particular sites such as around the coronary arteries and from either unenhanced scan for calcium scoring or angiographic imaging [110]. However, radiation exposure precludes measurements repeated in time to control treatment effects. Another limitation stems from long segmentation times required. CMR is considered the gold standard method for EAT quantification, validated even in animal models for research purposes [111]. Apart from being radiation-free, this technique offers potential multiple evaluations of underlying myocardial ischemia, storage diseases like amyloidosis, intramyocardial triacylglycerol accumulation, and diffuse myocardial fibrosis, in addition to the evaluation of systo-diastolic function and myocardial work [112]. The limitations are the higher cost and the lower spatial resolution and reproducibility than CCT, and eventual contraindication in some patients. To associate anatomical data and functional information, EAT may be evaluated on positron emission tomography/CCT [113]. Validation of semi-automatic or fully automated new methods is under way to make the CCT/CMR measurement of EAT easier and more suitable in the clinical setting [114].

## 7. Therapeutic Interventions Modifying EAT

Patients with HFpEF phenotype do not benefit from most standard HF therapies and, up to now, no specific treatment improving their survival has been identified. Benefit may be obtained through control of associated morbidities and risk factors, mainly by implementation of a healthy lifestyle including smoking cessation, weight reduction, exercise, and healthy diet [115]. In addition, the ESC 2016 guidelines recommended the symptomatic use of diuretics with a careful dose adjustment to avoid hypovolemia and severe preload reduction [116].

In recent years, the recognition of EAT as a crucial pathogenic factor for HFpEF has increased the interest towards therapeutic strategies that target this fat depot. The weight loss achieved by lifestyle modifications and by pharmacological and/or surgical interventions may have beneficial results. More specifically, treatments ameliorating the altered composition and distribution of fat stores may produce the best results [117]. Effects of glucose-lowering drugs on EAT in clinical studies are synthetized in Table 1.

### 7.1. Non-Pharmacologic Procedures Inducing Weight Loss

Studies evaluating the changes of EAT associated with various modalities and intensities of physical activity show ambiguous results. A recent systematic review and meta-analysis reports that exercise has no effect on BMI, but may reduce EAT and waist circumference [136]. In another meta-analysis analyzing only randomized controlled trials, the endurance training emerged as a particularly effective strategy for reducing epicardial fat depot [137]. In patients with moderate and severe obesity, a weight loss induced by low-calorie diets and exercise showed reductions of BMI, visceral fat and EAT [138]. Overall, epicardial fat is relatively resistant to dietary regimens determining modest weight loss. Instead, a very low-calorie diet in severely obese subjects was associated with a higher EAT shrink than overall adiposity, and a better correlation with cardiac morphological and functional changes such as LV mass reduction and diastolic function amelioration, indicating a cardio-protective effect. Interestingly, the change of EAT thickness was greater and faster than that of BMI and waist circumference [139]. In in vitro experiments, a 96 h low-glucose calorie restriction triggered changes in the secretome of cultured mature human adipocytes by reversing the expression of detrimental adipokines associated with metabolic complications [140]. Bariatric surgery was significantly associated with EAT shrinking in some but not all studies [141,142,143]. Kokkinos et al. report a greater reduction in EAT with Roux-en-Y procedure than sleeve gastroplasty, despite similar weight loss or reduction in waist circumference [144]. In a study using biliopancreatic diversion, a rapid reduction in SAT and visceral adipose tissue was observed, whereas EAT and PAT exhibited a significant decrease at the 12-month assessment [145]. A study comparing these three weight loss strategies reports that the relationship between change in BMI and change in EAT is significant for diet, less apparent for bariatric surgery, and virtually absent for exercise [146]. A recent systematic review with a meta-analysis of randomized and nonrandomized trials, reports that the shrinkage of epicardial fat adiposity reached with pharmacologic agents is more specific and clinically meaningful than with the above reported measures. Particularly, both lipid and glucose-lowering agents independently reduce EAT mass, although with a significant heterogeneity between studies [147].

### 7.2. Hypocholesterolemic Drugs

Several studies support a reduction of HFpEF morbidity and mortality induced by statins, a benefit that might depend in part on EAT and systemic inflammation reduction, and not on the effect on diabetes [148,149]. In a cohort study, atorvastatin and simvastatin/ezetimibe induced similar improvements in cholesterol profiles and a 7% mean decrease of EAT after six to eight-month follow-up, with an effect significantly more pronounced for atorvastatin [150]. In a more recent study, statins induced over time and independently of intensity of LDL-cholesterol lowering, a decrease in the attenuation of EAT at CCT considered a marker of inflammation, but a neutral effect on subcutaneous fat [151]. Accordingly, in a multi-center study comparing 87 aortic stenosis patients on statins to 106 not on statins, hypocholesterolemic treatment was significantly associated with reduction of EAT thickness. Interestingly, the in vitro examination in the same study showed a significantly higher anti-inflammatory effect on EAT compared to SAT, supporting the hypothesis of a direct action of these drugs on EAT metabolic activity and pro-inflammatory cytokine secretome (IL-1b, IL-1ra, IL-2, IL4, IL-6, IL-8, IL-12, basic Fibroblast Growth Factor (FGF), Eotaxin, Granulocyte-Colony Stimulating Factor (G-CSF), Interferon (IFN)-γ, Platelet-Derived Growth Factor (PDGF), TNF-α, and Vascular Endothelial Growth Factor (VEGF) [152]. The exact mechanism of this specific pleiotropic effect remains unknown. In this connection, epicardial overexpression of certain lipoprotein receptors such as LRP1 and VLDLR has been found in T2DM patients compared to non-diabetic subjects [153]. A study with pro-protein convertase subtilisin/kexin type 9 inhibitors, reports a remarkable 20% reduction in EAT thickness after 6 months of treatment [154].

### 7.3. Glucose-Lowering Drugs Used in T2DM

#### 7.3.1. Metformin

Metformin, the most widely prescribed drug to treat T2DM, especially when associated with obesity, may exert beneficial effects on body weight and fat composition, as well as extra metabolic ancillary effects that are clinically very interesting [155]. A study investigating weight loss and energy metabolism by this drug in rats and in diabetic and non-diabetic people, found a significant reduction of visceral fat independently of diabetes. Moreover, an up-regulation of fat oxidation-related enzyme in the liver, of UCP-1 in the brown adipose tissue, and of UCP-3 in the skeletal muscle were found to support a possible mechanism of enhancement of fat oxidation [156]. Treatment in patients with metformin, as well as with atorvastatin, pioglitazone, and exenatide, is accompanied by increase of circulating omentin-1 levels, an adipocytokine expressed in omental and epicardial fat that ameliorates insulin-sensitivity and is provided of anti-inflammatory and CV protective effects [157]. In a canine model of atrial fibrillation induced by rapid pacing, metformin was able to reverse the alteration in the adipokine profile of EAT [158]. Recent clinical observations testify to a beneficial effect of metformin on epicardial fat. A small prospective study demonstrated that a 3-month monotherapy produced a significant reduction of EAT thickness in 40 newly diagnosed T2DM patients, and a similar finding was observed in 30 obese children using biguanide for insulin resistance correction [118,119]. These results might account for the lower mortality in metformin-treated people with HFpEF reported in a systematic review and meta-regression analysis, and for the amelioration of cardiac functional parameters possibly observed in patients with mild forms of HF, mainly HFpEF, no history of CV events, and after a long-time chronic treatment [159]. Several other pieces of clinical evidence support a protective effect of metformin against heart failure [160]. Recently, two randomized trials evaluating EAT changes induced after six months by sitagliptin or liraglutide as add-on therapy in overweight/obese T2DM individuals inadequately controlled on metformin monotherapy showed that both incretin-based therapies significantly reduced EAT, whereas no such result was detected in the metformin groups [122,123]. These findings are indicative of the inferiority of metformin used alone, even in the absense of a comparison with placebo, but do not exclude a synergistic effect on EAT change with the added glucose-lowering drug. In this regard, a trial evaluating the efficacy of add-on ipragliflozin, a sodium-dependent glucose transporter-2 inhibitor, versus metformin in T2DM patients treated with sitaglipin, reported a mean percentage reduction in visceral adipose deposits significantly greater in the ipragliflozin group but not absent in the metformin group (−12.06% vs. −3.65%, *p* = 0.040) [161]. This doubtful evidence concerning metformin may be clarified only by a dedicated placebo-controlled trial with EAT measurement.

#### 7.3.2. Thiazolidinediones

The PPAR-γ has been implicated in the attenuation of inflammation and oxidative stress determined by adiponectin in LPS-induced inhibition of pre-adipocyte differentiation [162]. Moreover, a reduced expression of this receptor was associated with the phenotype conversion of epicardial fat from BAT to WAT during the progression of atherosclerosis [163]. Thiazolidinediones (pioglitazone and rosiglitazone) are PPAR-γ agonists that, administered to T2DM patients, may cause weight gain with a favorable shift in fat distribution from VAT to SAT [164]. When administered to obese fatty Zucker rats, rosiglitazone induced a significant up-regulation of PGC1-α, responsible of a rapid browning of EAT and of increased rates of lipid turnover [165]. The treatment with pioglitazone in CAD patients with T2DM or metabolic syndrome, was shown to increase PPARγ in SAT and to selectively reduce the expression of IL-1β, IL-1Ra and IL-10 in EAT, and of IL-10 in SAT, irrespective of other clinical variables (weight, BMI, waist, fasting blood glucose, A1C, HOMA-IR, ACE inhibitors, angiotensin receptor blockers, statins, aspirin, and metformin) [120]. In a similar study in patients with CAD and metabolic syndrome, pioglitazone alone or in combination with simvastatin, substantially reduced plasma inflammatory markers as well as EAT cytokines (IL-6, TNF-α, resistin, asymmetric dimethylarginine and metalloproteinase-9) production [121]. In a clinical study, treatment with pioglitazone in patients with T2DM without clinical CV disease improved both diastolic and systolic function assessed with CMR, and this amelioration was closely related to improved myocardial insulin sensitivity evaluated after euglycemic insulin clamp with PET [166]. Unfortunately, therapy with thiazolidinediones is associated with fluid retention and risk for HF development, especially in T2DM patients with underlying diastolic dysfunction [167]. This adverse effect depends on a potent anti-natriuretic action that upregulate sodium and water transport channels in human proximal tubule cells [168].

#### 7.3.3. Incretin-Based Therapy

The glucagon-like peptide-1 receptor agonists (GLP-1 RAs) (lixisenatide, liraglutide, semaglutide, exenatide, albiglutide, and dulaglutide) represent one of the two categories of drugs indicated for the T2DM treatment that act on the incretin system [169]. In recent years, seven trials have demonstrated their superiority or almost non-inferiority for CV outcomes, indicating advantageous cardio-protective effects [170]. Among these, a study with liraglutide demonstrated a reduction in LV filling pressure and an improvement in diastolic function, both parameters relevant for progression of diabetic cardiomyopathy and symptomatic HFpEF [171]. This result was confirmed by a systematic review [172]. Other experimental studies support an effect of fat mass reduction of GLP-1 RAs by increasing insulin sensitivity, lipolysis, and energy expenditure. Liraglutide was seen to promote pre-adipocytes differentiation and to inhibit FA synthase in adipose tissues from mice [173]. Earlier studies provided evidence that GLP-1R was present in adipose tissue with an increased mRNA and protein expressions in visceral adipose depots, and that it promoted pre-adipocyte differentiation and increased number of small adipocytes [174]. The consequent positive effect against insulin resistance and obesity suggested a potential relationship with amelioration of local insulin sensitivity in obese subjects [175]. Furthermore, a central injection of liraglutide in mice stimulated thermogenesis of BAT and browning of white adipocytes independently of nutrient intake [176]. Despite these interesting preclinical data, the results of clinical studies that evaluated the impact of GLP-1-RA on EAT are not unequivocal. One recent placebo-controlled trial study reports that liraglutide primarily reduced SAT, but failed to decrease visceral, hepatic, myocardial or epicardial fat [124]. In another, the intention-to-treat analysis did not reveal a significant effect of liraglutide on most of adipose tissue compartments including EAT [125]. Other concerns may arise from the lack of anti-inflammatory properties of GLP-1RA. Intra-cerebro-ventricular exendin-4 in rats induced weight loss but not reduction of leptin, while an increased expression of TNF-α, MCP-1 and factors involved in extracellular matrix deposition was observed in repeated biopsies of abdominal SAT from T2DM patients treated with liraglutide [177,178]. However, other authors underline that the cardio-metabolic advantage offered by liraglutide may be the substantial and rapid reduction of EAT thickness (29% at 3 months and 36% at 6 months), a result observed in a study with T2DM patients randomized to liraglutide on top of metformin versus metformin alone. In those remaining on metformin no change in EAT thickness was registered [123]. Previously, one study reported that a 12-week short term treatment either with exenatide or liraglutide, determined a mild yet significant reduction of EAT thickness, BMI, HbA1c and weight in a smaller group of T2DM patients without difference between the two GLP-1RAs [126]. Similarly, in another trial using RM, in obese patients with uncontrolled T2DM, exenatide reduced liver fat content and epicardial fat, not associated with a significant HbA1c reduction, but with a significant relation with weight loss when compared with reference treatment (sulphonylureas or repaglinide alone or in combination with metformin) [127]. More recently, a controlled parallel study in 80 subjects with T2DM and obesity demonstrated that the weekly administration of semaglutide or dulaglutide, two long-acting GLP-1 RAs, caused a substantial (of the order of 20%) and dose-dependent reduction in EAT thickness after 12 weeks, while there was no EAT reduction in the control group taking metformin alone. Moreover, a not significant HbA1c and BMI reduction was observed in all groups of treatment [128]. Examining RNA-sequencing data from EAT samples obtained during elective cardiothoracic surgery in 5 T2DM patients with CAD and 3 nondiabetic subjects, Iacobellis et al. demonstrated that human EAT expresses both GLP-1R and GLP-2R genes without differences related to co-morbid diabetes [179]. This finding supports the hypothesis of a direct action of GLP-1 analogues on EAT and suggests the use of EAT as therapeutic target for GLP-1 activation even in non-diabetic people.

The other incretin-based therapy is represented by the DPP-4 inhibitors. These drugs act by prolonging the half-life of endogenous GLP-1 through inhibition of its cleavage, and are recommended in T2DM patients without CV risk [180]. Even if DPP4 mRNA has been found to be more expressed by human adipocytes isolated from EAT compared with SAT, only a pilot study showed that sitagliptin substantially and rapidly reduced EAT in subjects with obesity and T2DM inadequately controlled by metformin monotherapy [122,181]. In addition, while DDP4 inhibition increased UCPs in BAT of diet-induced obese mice, other data indicate a dangerous potentiation of the stromal cell-derived factor-1, a stem cell chemokine which can promote inflammation, proliferative responses, and neovascularization [182,183]. These findings, together with others indicating that reduction of DPP4 activity was associated with cardiac fibrosis and inflammation and with impaired ventricular function in older diabetic rodents, might explain why in randomized placebo-controlled studies vildagliptin induced an undesirable remodeling of LV and saxagliptin increased the risk of HF in T2DM patients [184,185,186].

#### 7.3.4. Sodium-Glucose Cotransporter 2 Inhibitors

Selective sodium-glucose cotransporter 2 inhibitors (SGLT2-Is) are a class of glucose-lowering drugs that bind on the SGLT2 transporter in the proximal tubule of the kidney and increase glucose excretion via hindering its reabsorption. Large-scale CV outcome trials in patients with T2DM have suggested that SGLT2-Is are superior to a matching placebo in reducing primary and secondary hospitalization due to HF and CV death [187,188]. Several potential mechanisms of this benefit have been proposed, among which promotion of natriuresis and glycosuria leading to osmotic diuresis, elevation of plasma levels of ketone bodies improving cardiac energetics and efficiency, inhibition of the Na^+^/H^+^-exchanger reducing the intra-cardiomyocyte sodium overload, and others [189]. Although the urinary loss of calories by SGLT2-Is causes only modest decreases in body weight, these drugs are able to especially shrink the visceral fat depots in obese T2DM patients [190,191]. Particularly, studies performed with various SGLT2-Is showed reduction of EAT thickness or volume, suggesting a drug class effect. Amelioration of EAT inflammation and reduced secretion of deleterious adipokines could be associated with this effect. Three simultaneous small studies evaluated respectively canagliflozin, luseogliflozin, and ipragliflozin in T2DM patients with or without obesity. After a 3–6-month period of treatment, all registered a significant reduction of EAT measured by echocardiography or CRM, with a consistent HbA1c and BMI reduction. Moreover, a positive correlation was established between C-reactive protein and EAT mass modified by luseogliflozin [129,130,131]. A year later, these results were confirmed in another small study by Sato et al. In 40 T2DM patients with CAD, a 24-week therapy with dapagliflozin compared to conventional therapy, caused a significant decrease in EAT volume evaluated by CCT, with a significant positive correlation with circulating TNF-α level [132]. More recently, a 24-week, double-blind, placebo-controlled, randomized clinical trial, showed for the first time that the dapagliflozin addition to metformin monotherapy significantly and rapidly, already after 12 weeks, reduced ultrasound-measured EAT thickness in overweight/obese patients with T2DM. The magnitude of EAT reduction was indeed higher after 24 weeks of treatment and significantly greater than body weight change (20% vs. 8% at week 24, respectively). An EAT reduction of smaller extent was documented in the placebo plus metformin group [133]. The discrepancy between EAT reduction and weight loss, already observed in a study with liraglutide by the same investigator, may likely depend on the EAT property of being a small fat depot with fast metabolism [123]. To understand the mechanism behind the positive effects of dapagliflozin on EAT, a study by Diaz-Rodriguez et al. analyzed SGLT2 expression using real-time polymerase chain reaction Western blot and immunohistochemistry, from 52 ex vivo adipose samples obtained from patients undergoing heart surgery [192]. It was found that SGLT2 was expressed in EAT but absent or low in SAT, and that EAT SGLT2 mRNA expression was higher in men compared with women. The EAT samples treated with dapagliflozin showed increased glucose uptake, reduced secretion of pro-inflammatory chemokines, improved differentiation of adipocytes, and beneficial wound healing in endothelial cells compared to those treated with insulin. All these changes were congruent with a positive metabolic reset of EAT induced by SGLT2 inhibition [192]. Discordant results emerged from the recent EMPACEF study where a 12-week therapy with empagliflozin effectively reduced liver fat in high-fat-high-sucrose diet mice and in T2DM patients, but failed to change the amount of epicardial fat [134]. However, it cannot be excluded that a longer observation could have yielded different results. If it were not for these disappointing results with empagliflozin, the data set supports the conclusion that EAT undergoes a multifaceted remodeling after SGLT2 inhibition, a trend that could be considered a class effect responsible, at least in part, for the cardioprotective properties largely documented in experimental studies on animal models [193,194,195,196,197,198,199]. In a recent and interesting investigation in human and murine HFpEF myocardium, empagliflozin significantly suppressed the increased levels of ICAM-1, VCAM-1, TNF-α, and IL-6 and attenuated oxidative stress and eNOS-dependent PKGIα oxidation, leading to amelioration of cardiomyocyte stiffness [200]. In human trials, empagliflozin therapy was associated with ventricular and micro/macrovascular remodeling [201,202]. Overall, these favorable cardiac effects may explain why SGLT2-Is may prevent HF in the experimental setting and reduce in T2DM patients the risk of mortality and admission to hospital for HF more than GLP-1 RAs, being the latter more effective in reduction of non-fatal stroke [194,203,204]. With the limit of EF measurements at the time of the event and not at baseline, canagliflozin reduced the overall risk of HF events in the T2DM participants to the CANVAS program, with no clear difference between HFrEF and HFpEF [205]. In October 2021, the positive results were released of the EMPEROR-Preserved Clinical Trial involving patients with HFpEF, diabetics or not, treated with empagliflozin [206]. Particularly, the study stated that empagliflozin determined over a median of 26.2 months, a 21% lower risk of composite CV death and hospitalization for HF, a finding mainly related to a 29% lower risk of hospitalization for HF and independent of the presence of diabetes. An ongoing trial will determine the efficacy and safety of dapagliflozin in HF patients with preserved and mildly reduced EF [207].

#### 7.3.5. Insulin Therapy

Insulin is known to stimulate adipocyte differentiation and lipid accumulation of WAT, and to increase lipogenesis in epicardial fat [32,208]. On the other hand, insulin may promote cardiac fibrosis and exerts a powerful anti-natriuretic effect [209,210]. These properties may explain why therapeutic regimens that include insulin are associated with the highest incidence rates of HF [211]. Only a pilot study has evaluated the effects on EAT/PAT of a 24-week therapy with detemir or glargine in insulin-naïve inadequately controlled T2DM patients. Unexpectedly, a significant reduction of EAT thickness was observed with respect to baseline values for both insulin analogues [135]. Of note, the tight glycemic control achieved with insulin therapy, particularly in acute coronary syndromes, results in an improvement in both cardiac performance and patient CV outcomes [212,213].

### 7.4. Pericardiectomy

A pericardial resection may eliminate the external restraining effect due to an enlarged EAT on the right heart, which substantially contributes to elevation of LV filling pressure. This intervention involving a minimally invasive percutaneous procedure might represent a potential option to treat HFpEF patients with severe LV restriction. At the moment, the technique has been successfully tested in untrained dogs and in a pig model of diastolic dysfunction [214,215]. In a pilot study in humans, anterior pericardiotomy without extensive pericardial resection, substantially attenuated the increase in LV filling pressures that developed during volume loading in presence of risk factors for HFpEF and no pericardial disease [216]. A trial is currently testing the efficacy of this surgical intervention in HF [217].

### 7.5. Future Therapies

The anti-inflammatory treatment would be the more coherent pathophysiological therapy for dysfunctional EAT and HFpEF. To date, no study evaluated the effects on epicardial fat of such a strategy. Some data are available on biological agents administered to subjects with HFpEF. In a pilot study, the competitive IL-1 receptor antagonist anakinra significantly ameliorated after 14 days the systemic inflammation and the aerobic exercise capacity of patients with HFpEF and elevated plasma CRP levels [218]. However, the positive result on cardiorespiratory fitness was not confirmed in the follow-up phase II clinical trial, supposedly because of the very high prevalence of obesity [219]. Hopes for the development of effective drugs targeting the inflammatory secretome of EAT.

## 8. Conclusions

Diabetes is an important driver of HF, in particular of HF with the features of HFpEF in which context it represents, together with obesity, the phenotype associated with the poorer prognosis. The distinctive pathophysiology of HFpEF is focused on the central role of systemic inflammation associated with metabolic diseases, which is responsible of myocardial injury, but also of expansion, inflammation, and hypermetabolic activity of epicardial fat. The strict intimacy between this dysfunctional fat depot and the underlying cardiac muscle enables EAT to function as a booster of myocardial damage that exacerbates the microvascular dysfunction and fibrosis through paracrine and vasocrine effects. From this perspective, EAT may represent a key contributor to the development of HFpEF, and therefore a useful imaging biomarker of risk for CV adverse events that could be used to identify those patients who might benefit from EAT modifying therapies. To this end, it is desirable to achieve a robust method for depicting and quantifying the epicardial fat, that may be realistically and routinely usable in hospital wards and have low operator dependence. At the same time, EAT could be seen as an appealing therapeutic target to assess the efficacy of drugs that modulating epicardial adiposity may ameliorate, or better yet, prevent HFpEF. This is a crucial objective, considering that HFpEF at moment lacks effective treatments other than lifestyle optimization and symptomatic use of diuretics, as well as of studies demonstrating a therapeutic improvement in its prognosis. Metformin, GLP1-RAs and especially SGLT2-Is can be effective to alleviate EAT dysfunction. The concordance of their anti-inflammatory and anti-hyperglycemic benefits confers an added value to drugs that are currently used for T2DM but might be even valuable to manage HFpEF. The same applies to other anti-inflammatory medications such as statins that are very frequently part of the therapeutic equipment of the diabetic patients. Based on the most recent clinical evidence, a multifactorial therapeutic approach seems to be the most effective therapeutic strategy in reducing the risk of CV events in the diabetic patient [220]. To substantiate this approach, there is a need to fill knowledge gaps through further research. Primarily, it should be elucidated whether the manipulation of both structural and functional properties of EAT results in a better outcome for individuals with diabetes and HFpEF or other CV diseases. If so, it is necessary to better clarify the mode by which the current medicaments may attain the benefit and hopefully to find others targeted therapies perhaps more effective.

## Figures and Tables

**Figure 1 biomolecules-12-00176-f001:**
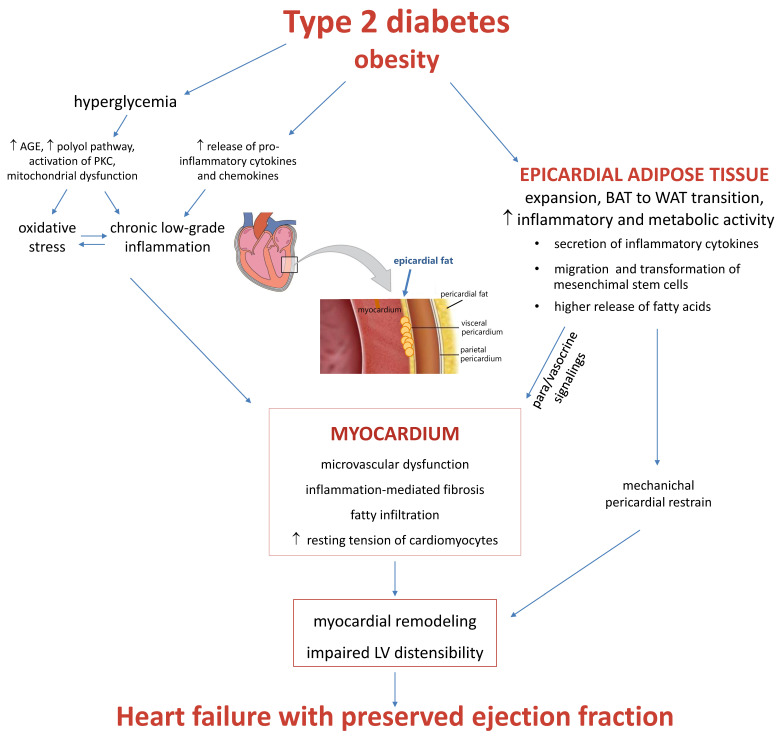
Two Inflammatory and microvascular mechanisms.

**Table 1 biomolecules-12-00176-t001:** Effects of glucose-lowering drugs on EAT in clinical studies. No reported study evaluated HFpEF or other CV outcomes.

	Type of Study	Type of Patients (*N*)	Treatment Dose	Treatment Duration	Clinical and Laboratory Parameters	Plasma or EAT Cytokines	EAT Mass Shrinking	References
Metformin	CCT	New diagnosed T2DM (40)	1000 mg bd	3 months	↓ BMI	Amelioration	Yes	[118]
	Observational	Obese Children (30)	dose NA	3 months	↓ BMI, ↓ BW, ↓ HOMA-IR	NA	Yes	[119]
Pioglitazone	EAT biopsy	T2DM with CAD (11)	25 mg (average)	24 months (average)	↓ expression of IL-16, IL-1Ra, and IL-10 in EAT	NA	NA	[120]
	CCT	Metabolic syndrome with CAD (36)	15/30 mg daily with or without simvastatin	3 months	NA	Amelioration	NA	[121]
Sitagliptin (DPP4-I)	Pilot	Obese T2DM (26)	50 mg + metformin 1000 bd vs. metformin 1000 mg bd alone	24 weeks	↓ BMI	NA	Yes	[122]
GLPI-RAs								
Liraglutide	RCT	T2DM (54)	1.8 mg s.c. daily	6 months	↓ HbA1c, ↓ BMI	NA	Yes	[123]
Liraglutide	RCT	T2DM (50)	1.8 mg s.c. daily	26 weeks	↓ BW	NA	No effect	[124]
Liraglutide	RCT	T2DM (47)	1.8 mg s.c. daily	26 weeks	↓ BW	NA	No effect	[125]
Exenatide/Liraglutide	CCT	T2DM (12/13)	5/10 mcg se bd/1.2 mg se daily	3 months	↓ BMI, ↓ BW, ↓ HbAlc	NA	Yes	[126]
Exenatide	RCT	Obese T2DM (44)	5/10 mcg bd	26 weeks	↓ BW	NA	Yes	[127]
Semaglutide/Dulaglutide	RCT	Obese T2DM (30/30)	1 mg sc weekly/1.5 mg sc weekly	12 weeks	↓ HbA1c, ↓ BMI	NA	Yes	[128]
SGLT2-Is								
Canagliflozin	Small study	T2DM (13)	100 mg	6 months	↓ HbAlc	NA	Yes	[129]
Luseogliflozin	Pilot	Overweight/ObeseT2DM (19)	2.5–5 mg	12 weeks	↓ HbA1c, ↓ BMI, ↓ HOMA-IR, ↓ BP	NA	Yes	[130]
Ipragliflozin	Pilot	T2DM (9)	50 mg	12 weeks	↓ HbA1c, ↓ BMI, ↓ HOMA-IR	Lleptin	Yes	[131]
Dapagliflozin	Small study	T2DM with CAD (40)	10 mg vs. conventional therapy	6 months	↓ HbAlc, ↓ BMI	TNF-a	Yes	[132]
Dapagliflozin	RCT	Overweight/obese T2DM (100)	10 mg + metformin vs. metformin alone	24 weeks	↓ HbA1c, ↓ BMI	NA	Yes	[133]
Empagliflozin	CCT	T2DM (56)	10 mg	12 weeks	↓ HbAlc, ↓ BW	NA	No effect	[134]
Detemir/Glargine	Pilot	T2DM (36/20)	10 UI (initial dose)	6 months	↓ HbAlc (Glargine)	NA	Yes	[135]

Abbreviations: Bd: bis in die; BP: blood pressure; BW: body weight; CAD: coronary artery disease; CCT: Controlled Clinical Trial; HOMA-IR: homeostatic model assessment of insulin resistance; NA: not available; RCT: Randomized Clinical Trial.

## Data Availability

Not applicable.

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
