# Peer review of "Dysregulated Epicardial Adipose Tissue as a Risk Factor and Potential Therapeutic Target of Heart Failure with Preserved Ejection Fraction in Diabetes"

_biomolecules, 2022, doi:10.3390/biom12020176_

Round 1

Reviewer 1 Report

This is an extremely well written and detailed review on epicardium fat pad as a risk factor for CHF.

the authors have clearly delineated some of the vital studies and their limitations.

it would have been wonderful to know if there is a certain percentage of EAT reduction that can avoid cardiac complications but the review is clear that the data thus far is equivocal.

I would like the authors to consider elaborating a little on the lines 132-135 eg where are the scretomes generated from etc.

Overall this is a very well laid out review article.

Author Response

We wish to thank the reviewer for the precious comment. Accordingly, we modified the text to better explain this point.

Reviewer 2 Report

This report y Salvatore et al is a review specifically on the importance of epicardial adipose tissue phenotype in diabetes and heart failure with preserved ejection fraction (HFpEF). Overall, the topic is scientifically well covered, however, some important comments/questions have to be raised.

A main issue is the lack of distinguishing between type 2 and type 1 diabetes in most of the report; therefore the title is misleading.

As this reviewer read the whole report, the title should rather be:

«Dysregulated (or dysfunctional) epicardial adipose tissue as a risk factor and potential therapeutic target of heart failure with preserved ejection fraction in diabetes”

1) The anatomic and functional properties of EAT (pt.2) is well written, displaying both the protective and harmful properties.

 2) In the description of EAT in diabetes (pt 3) it is especially visualized that most of the report is not precise regarding the type of diabetes; for instance – EAT is associated with insulin resistance and metabolic syndrome which is typical for type 2  diabetes etc.

3) Diabetes and HFpEF (pt 4), among the mentioned pathophysiological mechanisms discussed, any comment on the importance of glucose per se, probably the most important is lacking.

 4) In pt 5 very much repeating from pt 3. It is also imprecice in the presentation – i.e. the relationship HFpEF/ diabetes/obesity and EAT? Is it obesity or diabetes that is the main issue? Again, distinguishing between type 1 and type 2 should be done.

5) Very narrow selection on cytokines/inflammatory mediators referred to. 

6) The diagnostic tools available for EAT enlagement is well covered and discussed

7) The introductions/described mechanisms to the different treatment modalities is well written.

However, the main challenge is the description and discussion of treatment options, especially for pharmacological treatment. Herein there are animal studies, in vitro studies, human studies (observational, clinical randomized..) discussed together in a confusing format, without being precise on whether the effect are on diabetes (and type), HFpEF or EAT; or in combination. The table page 9 mainly refers to metanalyses and some randomized studies, are accompanied by a very broad text that includes and discuss a lot of other results. This table should be replaced by one (or several) tables that specifically show the type of study, type of patients (diabetes type, obese patients) in the clinical studies, treatment modlitiy/ies, duration of treatment, effects measures, effects on EAT, effects on HFpEF, all with the specific references. That would give the reader a better overview of the main issue – the treatment effects on EAT in diabetes patients with HFpEF. Although, with some added text.

8) The conclusion seems sound, although the potential treatment modalities could even more be presented as an indrect treatment og HFpEF. The importance of a valid, robust, realistic method for depicting and quantifying EAT is nicely focused. 

Author Response

This report y Salvatore et al is a review specifically on the importance of epicardial adipose tissue phenotype in diabetes and heart failure with preserved ejection fraction (HFpEF). Overall, the topic is scientifically well covered, however, some important comments/questions have to be raised.

A main issue is the lack of distinguishing between type 2 and type 1 diabetes in most of the report; therefore the title is misleading.

As this reviewer read the whole report, the title should rather be:

«Dysregulated (or dysfunctional) epicardial adipose tissue as a risk factor and potential therapeutic target of heart failure with preserved ejection fraction in diabetes”

We wish to thank the reviewer for the precious comment. Accordingly, we modified the title as requested.

  • The anatomic and functional properties of EAT (pt.2) is well written, displaying both the protective and harmful properties.

We wish to thank the reviewer for the comment.

  • In the description of EAT in diabetes (pt 3) it is especially visualized that most of the report is not precise regarding the type of diabetes; for instance – EAT is associated with insulin resistance and metabolic syndrome which is typical for type 2  diabetes etc.

We wish to thank the reviewer for the precious comment. Accordingly, we modified the text to better define the study populations.

  • Diabetes and HFpEF (pt 4), among the mentioned pathophysiological mechanisms discussed, any comment on the importance of glucose per se, probably the most important is lacking.

We wish to thank the reviewer for the precious comment. Accordingly, we modified the text to better explain this point.

  • In pt 5 very much repeating from pt 3. It is also imprecice in the presentation – i.e. the relationship HFpEF/ diabetes/obesity and EAT? Is it obesity or diabetes that is the main issue? Again, distinguishing between type 1 and type 2 should be done.

We wish to thank the reviewer for the precious comment. Accordingly, we modified the text to better explain this point. If mentioned in the studies selected as references, we have now specified whether the population was diabetic, obese or both, and whether it was type 1 or type 2 diabetes mellitus. As explained in pt 3, EAT is generally observed growing together with growing of BMI, both in diabetic and in non-diabetic patients. However, in pt5 we have now mentioned that, also in non-obese diabetic patients, EAT could be associated with HFpEF. Thus, we have now better specified the study populations, but due to the lack of broad data in literature, it’s hard to establish the main issue between diabetes and obesity. We also cut out the redundant parts.

  • Very narrow selection on cytokines/inflammatory mediators referred to.

We wish to thank the reviewer for the comment. Accordingly, we better specified which cytokines/inflammatory mediators are involved in pt 2.2, 7.2, 7.3.

  • The diagnostic tools available for EAT enlagement is well covered and discussed

             We wish to thank the reviewer for the comment.

7) The introductions/described mechanisms to the different treatment modalities is well written. However, the main challenge is the description and discussion of treatment options, especially for pharmacological treatment. Herein there are animal studies, in vitro studies, human studies (observational, clinical randomized...) discussed together in a confusing format, without being precise on whether the effect are on diabetes (and type), HFpEF or EAT; or in combination.

We wish to thank the reviewer for the comment. Accordingly, we modified the text to make more readable and less confused the paragraph, and to better specify the effect of drugs on EAT, diabetes and HFpEF. However, how emphasized in the text, there is a very poor literature about trials enrolling patients with diabetes and heart failure, in which the authors evaluated EAT.

The table page 9 mainly refers to metanalyses and some randomized studies, are accompanied by a very broad text that includes and discuss a lot of other results. This table should be replaced by one (or several) tables that specifically show the type of study, type of patients (diabetes type, obese patients) in the clinical studies, treatment modality/ies, duration of treatment, effects measures, effects on EAT, effects on HFpEF, all with the specific references. That would give the reader a better overview of the main issue – the treatment effects on EAT in diabetes patients with HFpEF. Although, with some added text.

We wish to thank the reviewer for the comment. Accordingly, we revised and improved the table by adding right references and other study features.

8) The conclusion seems sound, although the potential treatment modalities could even more be presented as an indrect treatment og HFpEF. The importance of a valid, robust, realistic method for depicting and quantifying EAT is nicely focused. 

We wish to thank the reviewer for the comment.

Reviewer 3 Report

This paper addresses epicardial fat in type 2 diabetes as a risk factor and therapeutic target of heart failure. The review is well organized and is interesting for those who work in this field. There are, however, some concerns that need to be addressed:

1-    Although I understand that the authors have work in this area, there is substantial reference to their own previous publications. This should be minimized.
2-    In chapter 2, it is reported that “despite whitening with age, it retains biological properties of BAT”. Is EAT classified as beige adipose tissue? And has both important thermogenic or endocrine functions? This should be clear.
3-    In line 96 it is stated that “EAT act as scavenger for excess FAs in systemic circulation to protect against their myocardial lipotoxicity responsible of functional deterioration”, but in line 135 it is referred that “high FAs feeding of heart by EAT may lead to intra myocardial fat accumulation causing functional derangements”. This is somewhat confusing: is lipid storage protective or deleterious, or both and in which conditions?
4-    Figure 1 could be improved, by adding a more appellative scheme and demonstrating the localization of PAT and EAT. The causes of “systemic inflammation and oxidative stress” should be added. A legend is also missing. Furthermore, there is no reference in the text to the figure.
5-    Table 1 must be revised and improved. Negative results should also be included (e.g. references 146 and 147 for metformin). And references must be revised (eg. references 134, 135 and 136, that appear related to metformin in the table, are not cited in the correspondent text - subchapter 7.3.1)
6-    Please introduce paragraphs in large Sections, such as 2, 4 and 5. Otherwise it is hard to read.
7-    Subchapters 7.3.3 and 7.3.4 should be shortened.
8-    Line 28: Correct “sodium-glucose transporter 2 (SGLT-2)” to “sodium-glucose cotransporter 2 inhibitors (SGLT2-Is)”.
9-    Line 195: Insert the meaning of “AGEs” and “LVEF”. Some readers may not be familiarized with these abbreviations.
10-    Line 201: Correct “abnormalitiesis”.
11-    Line 268: Correct “ciytokines”.

Author Response

This paper addresses epicardial fat in type 2 diabetes as a risk factor and therapeutic target of heart failure. The review is well organized and is interesting for those who work in this field. There are, however, some concerns that need to be addressed:

1-    Although I understand that the authors have work in this area, there is substantial reference to their own previous publications. This should be minimized.

We wish to thank the reviewer for the comment. The references are all consistent with the text and, accordingly with the comment, with our competence in the field. However, we have decided to eliminate some not strictly necessary references (137, 192 and 218).

2-    In chapter 2, it is reported that “despite whitening with age, it retains biological properties of BAT”. Is EAT classified as beige adipose tissue? And has both important thermogenic or endocrine functions? This should be clear.

We wish to thank the reviewer for the comment, accordingly we better specified the first point by adding a little period with reference [37]. For what concerns the second point, from “However, there is no direct evidence of heat production” to “paracrine fashion [24].”, we described that there is no clear evidence of a direct thermogenic function, even if EAT can releases thermogenic adipokines.

3-    In line 96 it is stated that “EAT act as scavenger for excess FAs in systemic circulation to protect against their myocardial lipotoxicity responsible of functional deterioration”, but in line 135 it is referred that “high FAs feeding of heart by EAT may lead to intra myocardial fat accumulation causing functional derangements. This is somewhat confusing: is lipid storage protective or deleterious, or both and in which conditions?

We wish to thank the reviewer for the comment. Accordingly we better clarified this point (ref [50]).

4-    Figure 1 could be improved, by adding a more appellative scheme and demonstrating the localization of PAT and EAT. The causes of “systemic inflammation and oxidative stress” should be added. A legend is also missing. Furthermore, there is no reference in the text to the figure.

We wish to thank the reviewer for his/her comment, accordingly we improved the figure. Now the localization of PAT has been shown and for this reason, a little period with reference, which describes how PAT and EAT can be used in confusing manner, has been added [23]. Moreover, following the suggestion of reviewer, we added the causes of systemic inflammation and oxidative stress.

5-    Table 1 must be revised and improved. Negative results should also be included (e.g. references 146 and 147 for metformin). And references must be revised (eg. references 134, 135 and 136, that appear related to metformin in the table, are not cited in the correspondent text - subchapter 7.3.1)

We wish to thank the reviewer for the comment. Accordingly, we revised and improved the table by adding right references and other clinical study features.

6-    Please introduce paragraphs in large Sections, such as 2, 4 and 5. Otherwise it is hard to read.

We wish to thank the reviewer for the comment, accordingly we introduced sub-paragraphs to render more readable the text.

7-    Subchapters 7.3.3 and 7.3.4 should be shortened.

We wish to thank the reviewer for the comment. Accordingly, we shortened the paragraphs. 

8 -    Line 28: Correct “sodium-glucose transporter 2 (SGLT-2)” to “sodium-glucose cotransporter 2 inhibitors (SGLT2-Is)”.

We wish to thank the reviewer for the comment, accordingly we corrected the word.

9-    Line 195: Insert the meaning of “AGEs” and “LVEF”. Some readers may not be familiarized with these abbreviations.

We wish to thank the reviewer for the comment, accordingly we specified these abbreviations.

10-    Line 201: Correct “abnormalitiesis”.

We wish to thank the reviewer for the comment, accordingly we corrected the word.

11-    Line 268: Correct “ciytokines”.

We wish to thank the reviewer for the comment, accordingly we corrected the word.

Round 2

Reviewer 2 Report

The revision is satisfactory, and I have no further comments
